# Mediation Effect of Musculoskeletal Pain on Burnout: Sex-Related Differences

**DOI:** 10.3390/ijerph191912872

**Published:** 2022-10-08

**Authors:** Yong-Hsin Chen, Chih-Jung Yeh, Chun-Ming Lee, Gwo-Ping Jong

**Affiliations:** 1Department of Public Health, Chung Shan Medical University, Taichung 402, Taiwan; 2Department of Occupational Safety and Health, Chung Shan Medical University Hospital, Taichung 402, Taiwan; 3Department of Internal Medicine, St. Joseph’s Hospital, Yunlin 632, Taiwan; 4Department of Internal Medicine, Chung Shan Medical University Hospital, Chung Shan Medical University, Taichung 402, Taiwan

**Keywords:** burnout, musculoskeletal pain, sex-related differences, women, mediating factor

## Abstract

Burnout occurs when people are exposed to emotionally demanding work situations over an extended period, resulting in physical, emotional, and mental exhaustion. This study adopted the personal burnout (PB) and work-related burnout (WB) scales of the Copenhagen Burnout Inventory to measure burnout levels. Musculoskeletal (MS) pain is one factor influencing burnout. Previous studies have demonstrated that male and female hormones may contribute to sex-related differences in MS pain. This was an observational and cross-sectional study conducted at a medical-university-affiliated hospital in Taichung, Taiwan, in 2021. Data were collected for demographic characteristics, family structure, living habits, occupation, physical health, Nordic Musculoskeletal Questionnaire score, and Copenhagen Burnout Inventory score. Of the 2531 questionnaires that were distributed, 1615 (63.81%) valid questionnaires remained after those with missing data were excluded. The results demonstrated neck and shoulder pain (NBSP) is commonly associated with burnout among healthcare workers, with a higher prevalence among women than among men. With confounders being controlled for, women were found to experience substantially higher levels of personal and work-related burnout than men did. NBSP is a mediating factor; therefore, it is a key reason why women are more likely than men to experience burnout.

## 1. Introduction

Burnout is a state of physical, emotional, and mental exhaustion that results from long-term involvement in work situations that are emotionally demanding [1]. The development of burnout can be divided into five stages [2], the first stage is the enthusiasm period, in which people kept enthusiasm for work. The second stage is the so-called stagnation period, as people begin to suffer from a series of work pressures which causes decreasing enthusiasm, and the burnout risks to become initiated. The third stage is the frustration period, beginning to gradually form the so-called chronic stress which causes people to gradually lose their enthusiasm for work. The fourth stage is the apathy stage, and it results from no obvious returns and work powerlessness. The fifth stage is the intervention period, at which time habitual burnout will lead to physical and emotional problems, and people will start to seek help and intervention.

The clinical symptom included emotional exhaustion, physical fatigue, cognitive impairments, disturbed sleep, and functional impairment [3,4] and often resulted from combination stressors caused by long-term work and nonwork on individuals [5], and that finally led to sleep disturbances, even depression, or anxiety disorders [3]. Burnout affects approximately half of all nurses, physicians, and other clinicians [6], which not only influences the individual’s mental health but also results in high physician turnover and reduced clinical hours, causing losses totaling approximately 4.6 billion in the United States annually [7]. Notably, these factors influence both personal life and work and could reduce patient quality of care [8]. Moreover, the reasons for sustaining burnout are extensive and complex. Overtime [9], shift work [10], lack of sleep [10,11], and chronic diseases [12] are the primary reasons. In addition, work experience [13], good exercise habits [11], marriage, and parenthood [14] could be helpful for decreasing burnout levels.

The common musculoskeletal pain sites were the lower back, shoulder, and neck [15], and that could be associated with long work hours [16], occupational stress [17], alcohol consumption [18], sleep duration [19], and chronic diseases [20], respectively. In line reported, the neck pain or stiff shoulders, low back pain were most common in Japan workers; among them, the annualized costs of presenteeism per capita were over $400 due to pain. [21].

Both male and female sex hormones may contribute to the marked sex-related differences in the occurrence of certain musculoskeletal (MS) pain conditions [22]. Some studies for epidemiology have found that pain is more prevalent in women than in men [23], especially neck pain [24,25]. Coincidentally, a correlation was found between burnout and musculoskeletal (MS) pain in seemingly healthy individuals [26,27], too.

Based on the associations among women, MS pain, and burnout, whether women sustain high burnout than men due to MS pain was a noteworthy and interesting topic. Therefore, this study proposed two hypotheses to further explore: that sex differences in burnout (1) exist and (2) result from MS pain differences between women and men. Hopefully the results of this research can encourage the medical institutions to make sex-differentiated strategies of burnout to further reduce burnout, especially among female staff members.

## 2. Methods

This study adopted an observational and cross-sectional design, which was conducted at a medical-university-affiliated hospital in Taichung, Taiwan, in 2021. All 2531 medical staff who have served for one year in the hospital were distributed a QR code of google forms linked questionnaires by email, in which 1633 (64.52%) were returned, and 1615 (63.81%) were determined to be valid after those with missing data were excluded. The professional fields and sex of 1615 participants were in Appendix A Appendix A; those were reclassified as Physicians, Nurses, Professional and technical personnel, and Administration staffs. The questionnaires inquired into the participant’s basic demographic characteristics, family structure, living habits, occupation, and physical health; they were also used to collect data on the Nordic Musculoskeletal Questionnaire and Copenhagen Burnout Inventory.

The response options for level of education were “below high school”, “Bachelor’s degree”, “Master’s degree”, and “PhD”. The response options for marital status were “married” and “other”. Participants were also asked whether they were involved in raising children. The response options were “without child”, “one child”, “two children”, “three children”, and “over three children”. Raising at least one child was reclassified into a new variable, “parenthood”. The question of whether participants engage in leisure activities with family or friends (LAFF) during vacation time was also included in the questionnaire. The options, “always”, “often”, “sometimes”, “seldom”, and “never”, were scored as 100, 75, 50, 25, and 0 points, respectively. An item related to the presence of a listed chronic diseases was included in the questionnaire, and the presence of one or more diseases was classified as a “yes” response. Regarding smoking during the past month, responses of “never” or “have quit smoking” were classified as “no”. Regarding drinking during the past month, responses of “seldom” or “every day” were classified as “engaged in alcohol use at least once” (which was termed “ever AU”), whereas the option of “never” was classified as “no alcohol use ever”. Sleeping duration was classified as “<5 h”, “5–6 h”, “6–7 h”, “7–8 h”, or “>8 h”. These responses were reclassified as “sleeping duration <6 h” and “sleeping duration >6 h”. The participants were asked whether they exercised “at least once a day”, “at least once a week”, “at least once a month”, “less than once a month”, or “never”. Responses of “at least once a day” or “at least once a week” were reclassified as “regular exercise weekly”. For the question pertaining to overtime work, the options were “seldom”, “less than 45 h per month”, “45–80 h per month”, and “more than 80 h per month”. Because the participants seldom worked overtime work for more than 80 h, the responses were reclassified as “seldom overtime work” and “experiencing overtime work” (including less than 45 h, 45–80 h, and more than 80 h per month). The options for questions on work schedule were “day shift work”, “night shift work”, “irregular shift work”, and “regular shift work”.

The Nordic Musculoskeletal Questionnaire was developed by the Nordic Council of Ministers [28]. The aim was to develop and test a standardized questionnaire methodology allowing assessment pain of low back, neck, shoulder, and general complaints. The Nordic Musculoskeletal Questionnaire had been determined which is repeatable, sensitive, and useful screening and surveillance tool on pain [29,30] and the reliability was good (kappa values were between 0.51 and 0.68) [31]. Validity tested against clinical history and the Nordic Musculoskeletal Questionnaire found less than 20% disagreement [28]. This study employed a modified version of the Nordic Musculoskeletal Questionnaire that was translated into Mandarin Chinese by the Taiwan Institute of Occupational Safety and Health, which had been used popularly for research in Taiwan [32,33,34]. Questions focused on the presence of work-related pain during the preceding year at the following sites: neck; left shoulder; right shoulder; upper back; waist or lower back; left elbow; right elbow; left wrist; right wrist; left hip, thigh, or buttock; right hip, thigh, or buttock; left knee; right knee; left ankle; and right ankle. If a participant answered “yes” to any question regarding presence of work-related pain over the past year, they were also asked about the frequency of pain, which were every day, once a week, once a month, once every half year, or at least once a year; these responses were scored as 100, 80, 60, 40, and 20 points, respectively.

The Copenhagen Burnout Inventory was developed by researchers from Denmark [35]. In the Copenhagen Burnout Inventory, exhaustion is considered to be the core of the concept of burnout. The Copenhagen Burnout Inventory has been used to develop three scales, the personal burnout (PB) scale, work-related burnout (WB) scale, and client burnout scale, that can be applied separately to measure burnout in different settings (not just the service professions). The questionnaires had very high internal reliability (the Cronbach’s alphas were between 0.85 and 0.87) and were formulated to be understandable and accessible to all people [35]. The present study used the Chinese version of the Copenhagen Burnout Inventory [36], which is considered a reliable and valid tool for assessment of burnout. For the purposes of this study, only the PB and WB scales were used. The first six questions, which pertained to PB, were as follows:
“How often do you feel tired?”“How often are you physically exhausted?”“How often are you emotionally exhausted?”“How often do you think ‘I can’t take it anymore’?”“How often do you feel worn out?”“How often do you feel weak and susceptible to illness?”

Items 7–13, which concerned WB, were as follows:
7.“Is your work emotionally exhausting?”8.“Do you feel burnt out because of your work?”9.“Does your work frustrate you?”10.“Do you feel worn out at the end of the working day?”11.“Are you exhausted in the morning at the thought of another day at work?”12.“Do you feel that every working hour is tiring for you?”13.“Do you have enough energy for family and friends during leisure time?”

The response options—“always”, “often”, “sometimes”, “seldom”, and ”never/almost never”—were scored as 100, 75, 50, 25, and 0, respectively, with the exception of item 13, which was inversely scored (i.e., the responses were scored by minimum “always” = 0 and maximum “never/almost never” = 100, sequentially). Levels of PB and WB were represented by the mean of the total PB and WB scores (the sum of scores for items 1–6 and items 7–13), respectively.

Regarding the procedures of statistical analyses, we adopted four steps to determined women effect on burnout and their mediating factors.

Step 1: Factor analysis [37] was conducted on the Nordic Musculoskeletal Questionnaire results to determine the underlying variables that explained most of the responses.

Step 2: The present research adopted chi-square test/Fisher exact test (categorical variables) and *t* test or one-way ANOVA (continuous variables) to determine if the difference between survey variable and sex or PB/WB is significant in statistic. These survey variables were significant in statistic that would be as possible mediating factors between sex and burnout or risk/protective factors of burnout.

Step 3: The risk or protective factors of burnout would be added in linear regression models of women effect on burnout to determine whether controlling for covariates significantly affected associations of the independent variables (IVs) with dependent variable (DV).

Step 4: The mediation effect would be tested on the basis of the strategy proposed by Baron and Kenny [38], in which (1) the IV significantly affects the mediator (*a*, first-stage effect), (2) the IV significantly affects the DV in the absence of the mediator, (3) the mediator has a significant unique effect on the DV (*b*, second-stage effect), and (4) the effect of the IV on the DV (*c*′, direct effect) weakens upon addition of a mediator to the model. Among these strategies, item (2) is recommended but not required [39]. The mediation effect exists if the combination effect of first-stage effect and second-stage effect is stronger than direct effect (a×b>c’). A mediation model suitable for the combination of categorical and continuous variables was developed by Iacobucci in 2012 [40]; the formulas are as follows: $$\hat{Y}=b_{01}+cX$$
$$\hat{M}=b_{02}+aX$$
$$\hat{Y}=b_{03}+c^{\prime }X+bM$$
$$Z_{a}=\hat{a}/\hat{S_{a}}$$
$$Z_{b}=\hat{b}/\hat{S_{b}}$$
where *X* is an IV; $\hat{Y}$ is a DV; $\hat{M}$ is the adjusted variable (i.e., the mediating factor); *a* is a logistic or linear regression coefficient of *X* against $\hat{M}$ when $\hat{M}$ and *X* are a DV and an IV, respectively; *b* is the logistic or linear regression coefficient of $\hat{M}$ against $\hat{Y}$; *c* is the logistic or linear regression coefficient of *X* against $\hat{Y}$; and *c*′ is the logistic or linear regression coefficient of *X* against $\hat{Y}$ with $\hat{M}$ as the adjusting variable. The standard errors of *a* and *b* are represented by *s_a_* and *s_b_*, respectively.

If $\hat{M}$ and $\hat{Y}$ are all continuous variables, the original formula of the Sobel test can be applied; this formula is as follows.
$$Z=\frac{a\times b}{\sqrt{b^{2}s_{a}{}^{2}+a^{2}s_{b}{}^{2}}}$$

If $\hat{M}$ and $\hat{Y}$ are categorical variables or a combination of categorical and continuous variables, the original formula of the Sobel test is rederived into a new formula as follows.
$$Z_{mediation\;}\left(Z_{m}\right)=\frac{\frac{a}{s_{a}}\times \frac{b}{s_{b}}}{\sqrt{Z_{a}{}^{2}+Z_{b}{}^{2}+1}}$$

The results exceeding |1.96|, |2.57|, and |3.90| (for a two-tailed test) are significant at α = 0.05, 0.01, and 0.0001, respectively. 

Analyses were performed using SAS Enterprise Guide 7.1 software (SAS Institute Inc., Cary, NC, USA), and significance was set at *p* < 0.05.

## 3. Results

First, the research must determine the underlying variables that could maximum explain the Nordic Musculoskeletal Questionnaire. As indicated in Table 1, the prevalence of MS pain during the previous year for both shoulders, neck, waist or lower back, and upper back were 43.09%, 36.22%, 27.93%, and 16.90%, respectively. The mean scores for frequency of neck, waist or lower back, right shoulders, left shoulder, and upper back pain were 26.76 ± 37.64, 20.20 ± 34.72, 17.64 ± 33.89, 15.07 ± 31.62, and 12.90 ± 29.77, respectively. According to the principle proposed by Hair and Anderson (1995), [37] factors 1 and 2 were retained because their vector values exceeded 1. Although the eigenvalue of factor 3 was lower than 1, it was retained to maximize the explanatory power for the questionnaire. The factor loadings were converted into standardized scoring coefficients through varimax rotation. The relatively large factor loading values in bold for factors 1, 2, and 3 correspond to frequency scores for neck and both shoulders pain (NBSP), both ankles pain (BAP), and both knees pain (BKP), respectively.

Second, the present study would determine the sex-related difference on survey variables for determining possible mediating factors. As indicated in Table 2, men and women significantly differed in marital status *(p* = 0.003), parenthood *(p* = 0.002), weekly exercise habits *(p* < 0.0001), monthly alcohol use (*p* < 0.0001), level of education *(p* < 0.0001), shift schedule *(p* = 0.029), and profession *(p* < 0.0001). The proportions for marriage (56.15% vs. 46.42%), parenthood (51.50% vs. 41.70%), regular exercise weekly (70.10% vs. 54.95%), ever alcohol use (52.16% vs. 34.40%), master’s degree or above (28.57% vs. 16.06%), DS work (72.09% vs. 64.23%), and profession of physician (30.56% vs. 3.50%) were higher among men than women. As noted in Table 3, women experienced higher NBSP than men (0.04 ± 0.93 vs. −0.17 ± 0.84; *p* = 0.000).

Confounders of burnout were next identified and used as control variables in a multiple linear regression model of women’s effect on burnout. Table 2 and Table 3 indicate that ever alcohol use (mean = 38.47 ± 17.68; *p* < 0.0001), sleeping duration < 6 h (mean = 41.07 ± 18.84; *p* < 0.0001), experience overtime work (mean = 42.40 ± 18.09; *p* < 0.0001), presence of chronic diseases (mean = 38.92 ± 18.14; *p* < 0.0001), NBSP (β = 8.25; *p* < 0.0001), BAP (β = 1.46; *p* < 0.01), and BKP (β = 1.98; *p* < 0.01) were risk factors for PB. PB was significantly correlated with shift schedule *(p* < 0.0001) and profession *(p* < 0.0001). Regular exercise weekly (mean = 31.85 ± 15.95; *p* < 0.0001), LAFF (β = −0.12; *p* < 0.0001), and work experience (β = −0.17; *p* < 0.01) were protective factors for PB.

As shown in Table 3, ever alcohol use (mean = 36.40 ± 15.67; *p* < 0.0001), sleeping duration < 6 h (mean = 38.05 ± 17.01; *p* < 0.0001), master’s degree or above (mean = 32.41 ± 15.79; *p* < 0.05), experience overtime work (mean = 40.01 ± 15.71; *p* < 0.0001), presence of presence of chronic diseases (mean = 36.31 ± 16.57; *p* < 0.0001), NBSP (β = 6.32; *p* < 0.001), BAP (β = 1.42; *p* < 0.01), and BKP (β = 1.31; *p* < 0.05) were risk factors for WB. WB was significantly correlated with shift schedule *(p <* 0.0001) and profession *(p <* 0.0001). Marriage (mean = 31.99 ± 15.38; *p* < 0.0001), parenthood (mean = 31.47 ± 15.83; *p* < 0.0001), LAFF (β = −0.14; *p* < 0.0001), and work experience (β = −0.23; *p* < 0.0001) were protective factors for WB.

Table 4 illustrates the effect of being a woman on burnout in linear regression models M_0_, M_1_, M_2_, M_3_, M_4_, and M_5_. Men and women did not significantly differ in PB and WB when no adjustments were made for any variable (M_0_). The effect of being a woman was significantly correlated with PB (β = 2.56; *p* < 0.05) and WB (β = 2.52; *p* < 0.05) in model M_1_, which had controls for weekly exercise, monthly alcohol use, sleep duration per day, level of education, overtime work per month, shift schedules, profession, presence of chronic diseases, LAFF, and work experience. The effect of being a woman was still significantly correlated with PB (β = 2.83 and 2.86; *p* < 0.05 for both) and WB (β = 2.43 and 2.34; *p* < 0.05 for both) in the M_2_ (M_1_ controlled for marriage) and M_3_ (M_1_ controlled for parenthood) models, respectively. The effect of being a woman was significantly correlated with PB (β = 2.56; *p* < 0.05) and WB (β = 2.54; *p* < 0.05) in the M_4_ model (M_1_ controlled for BAP and BKP); however, according to M_5_ model, the residual effect of being a women for PB (β = 0.70 1.12; *p* > 0.05) and WB (β = 1.12; *p* 005) could be fully explained by NBSP, respectively.

Mediation effect of NBSP that women sustain high burnout would be further identified by mediation models. As shown in Table 5, the first stage effect (*a* = 0.21; *p* < 0.01) for women to NBSP was statistically significant. The second stage effects (*b* = 8.25, 6.30; *p* < 0.0001 for both) for NBSP to PB and WB were also statistically significant. According to the Sobel test results, NBSP mediated (*Z* = 3.44, 3.41; *p* < 0.01 for both) the relationships of gender with PB and WB. 

These results verify the two hypotheses of this study: sex differences in burnout exist, and these differences are the result of MS pain differences between women and men.

## 4. Discussion

First, we must review our findings and compare with the past study’s results to further confirm the two hypotheses proposed in the introduction. Burnout can be influenced by family, living habits, sleep, work-related factors, and health problems. For example, family members and friends play a vital role in preventing the development of burnout [41], and health workers reported that they could minimize burnout by getting support from their family [42]. The present study discovered that participants who were married or parents recorded lower levels of WB than participants who were neither married nor parents (31.99 ± 15.38 vs. 36.29 ± 16.76; 31.47 ± 15.83 vs. 36.33 ± 16.25; *p* < 0.0001 for both, Table 2). This indicates that family relationships can effectively relieve burnout from work, which is consistent with the results of previous studies.

A positive dose-response relationship was noted between physical activity and emotional well-being [43]. In addition, physical activity and healthy exercise habits have been shown to be effective for reducing the risk of burnout [44]. The present study observed that participants who engaged in regular exercise weekly reported noticeably lower levels of PB and WB compared with participants who did not exercise (33.43 ± 17.18 vs. 39.73 ± 18.59; 31.85 ± 15.95 vs. 37.45 ± 16.10; *p* < 0.0001 for both); this is consistent with the results of previous studies.

Whether alcohol reduces stress is debatable [45]. Previous studies have reported that burnout was strongly associated with alcohol abuse or dependence among American surgeons [46] and that burnout was significantly and positively correlated with higher AU among doctors, nurses, and residents [47]. In summary, alcohol use was closely associated with high burnout. Table 2 reveals that participants who consumed alcohol at least once a month sustained significantly higher levels for PB and WB than those who did not (38.47 ± 17.68 vs. 34.65 ± 18.13; 36.40 ± 15.67 vs. 32.89 ± 16.45; *p* < 0.0001 for both), which demonstrates that alcohol use is an unfavorable factor for burnout and does not effectively reduce burnout.

The development of burnout is closely related to lack of sleep (<6 h) [33,48] and disturbed sleep [49]. For example, nurses who slept less than 6 h per working day had higher risks of PB (odds ratio = 3.0; *p* < 0.05) and WB (odds ratio = 3.4; *p* < 0.05) than those who slept more than 7 h per working day [50]. Table 3 illustrates that participants whose sleep duration was less than 6 h reported higher levels of PB and WB than those whose sleep duration was more than 6 hours (41.07 ± 18.84 vs. 32.94 ± 16.79; 38.05 ± 17.01 vs. 31.78 ± 15.26; *p* < 0.0001 for both). This is consistent with the results of previous studies.

Overtime work hours were significantly correlated with burnout in a dose-dependent manner [51]. Irregular shift (IRS) working schedule was also related to a significantly higher burnout rate [52]. The results of the present study are consistent with those of previous studies reporting high levels of PB and WB in participants who experienced overtime work or IRS at least once a month *(p* < 0.0001 for all, Table 2).

The prevalence of diseases is related to the severity of burnout. Among individuals with at least one physical illness, approximately half (54%) had no burnout, whereas 63% had mild burnout and 71% had severe burnout [53]. Moreover, burnout has been identified as an independent risk factor for future incidence of specific chronic diseases such as coronary heart disease [54] and type 2 diabetes [55]. This study had similar findings (Table 2); specifically, participants who had at least one presence of chronic diseases reported higher levels of PB and WB than those without presence of chronic diseases (38.92 ± 18.14 vs. 34.24 ± 17.75; 36.31 ± 16.57 vs. 32.84 ± 15.89; *p* < 0.0001 for both).

For people with high stress levels, participating in leisure activities can relieve stress, improve one’s emotional well-being, and maintain one’s physical and mental health [56,57,58]. Table 3 indicates that the LAFF was negatively correlated with PB (β = −0.12; *p* < 0.0001) and WB (β = −0.14; *p* < 0.0001). This suggests that LAFF is a protective factor for burnout, which is consistent with the findings of previous studies.

The effect of being a woman on the likelihood of burnout was not significant in the unary linear regression model (M_0_). Notably, there were significant difference on sex and PB or WB among marriage state, parenthood, exercise habit weekly, alcohol use habit in a month, education degree, shift schedule, and medical professional. This difference could interfere women effect on burnout. Therefore, we adopted the multiple linear regression models to control these confounders for determining if exist women effect on burnout. A significant correlation between women and burnout was really found in M_1_ (β = 2.56/2.52, *p* < 0.05 for both), M_2_ (β = 2.83/2.43, *p* < 0.05 for both), M_3_ (β = 2.86/2.34, *p* < 0.05 for both), and M_4_ (β = 2.56/2.54, *p* < 0.05 for both) model. This finding determined our hypothesis 1: sex differences in burnout really exist. In addition, M_3_ and M_4_ reflected the explanatory power of marriage and parenthood on the residual effect of being a woman on burnout. The results in Table 4 demonstrate that changing the β value from 2.56/2.52 to 2.86/2.34 still resulted in statistical significance, indicating that the influence of marriage and parenthood on the residual effect of being a woman on burnout were not obvious. Although support from family [42] and disturbance in home and family life [59] were associated with the development of burnout, this effect did not significantly differ between men and women. This hints that sex differences in burnout do not result from family structure.

Despite the musculoskeletal system is complex, the sites on musculoskeletal pain still could be suitably classified by important anatomical locations or scales such as the Nordic Musculoskeletal Questionnaire. Regarding the occurrence of pain, the past study demonstrated the lower back (26.9%), shoulders (20.9%), and neck (20.6%) were the most reported pain sites based on point prevalence [15]. We adopted the Nordic Musculoskeletal Questionnaire to survey the occurrence site and frequency on MS pain. However, we will face a problem that the survey results could be too complex on MS pain. Fortunately, factor analysis method helps us to overcome this difficult and gain a satisfactory result. Table 1 indicates three most explain pain’s sites on the questionnaire from high to low are neck and both shoulders (factor 1), both ankles (factor 2), and both knees (factor 3) through factor analysis, respectively. The past studies suggested neck or shoulder pain could be associated with mental health. For instance, one meta-analysis reported that the job strain was associated with the risk of low back pain, neck pain, shoulder pain and back pain [60]. Another study also reported that baseline levels of burnout predicted the onset of regional neck, shoulder, and lower back pain [27]. Our study’s result determined the pain sites on neck and shoulders really are strongly associated with mental health (such as burnout) than other pain sites. The specific description is shown in Table 3; PB and WB were correlated with NBSP (β = 8.25; 6.32; *p* < 0.0001 for both), BAP (β = 1.46; 1.42; *p* < 0.01 for both), and BKP (β = 1.98; *p* < 0.01; β = 1.31; *p* < 0.05). Among them, NBSP was more influential on the likelihood of burnout than other sources of MS pain.

Compared with men, women often reported neck, shoulders, waist, or back pain [23,24,25,61]. From a physiological viewpoint, sex-related differences in pain could result from estrogen and progesterone (the major female sex hormones) [22]. In addition, testosterone, the major male sex hormone, protects men from these chronic MS pain conditions [22]. These research studies suggested sex difference on MS pain exist. Our finding also confirms above result. Table 3 presents data showing that women sustained higher levels of NBSP than men (women vs. men: 0.04 ± 0.93 vs. −0.17 ± 0.84; *p* < 0.01). In addition, M_1_, M_2_, M_3_, and M_4_ models in Table 4 have determined women really sustain high level on burnout than men in adjusting confounders. Based on the above analysis, sex difference really exists on NBSP and burnout. To explore NBSP effect on the relationship between sex and burnout, we added NBSP to multiple linear regression model in Table 4 (M_4_) and was as a new model M_5_. The new model indicated that NBSP could fully explain (β = 0.70, 1.12; *p* > 0.05 for both) the residual effect of being a woman on burnout after all confounders were controlled for. This suggests that women who reported high levels of burnout may also sustain high NBSP. Based on this, we come back the beginning second hypothesis in Introduction: sex difference in burnout results from MS pain differences between women and men. We adopted mediation model (Table 5) to test this hypothesis. The result determined NBSP mediated (*Z* = 3.44, 3.41; *p* < 0.01 for both) the relationships of gender (specifically, being a woman) with PB and WB, which further indicates that NBSP primary explains why women sustain high levels of burnout. According to the two results, we could determine women easily suffer from neck and both shoulders pain than men, that would be important reason that women sustain high PB and WB than men. To the best of our knowledge, this finding has rarely been mentioned in the literature. Except to this, another value on the present study is proposing an effectively strategy to medical institutions that the improving plan on burnout should more focus on the MS pain problem and supply the more accessibility resource on relieve pain for women.

The women ratio is 81.26% in all 2531 distributed questionnaires. Among them, women proportion is higher than men in valid questionnaires (81.36% vs. 18.64%), too. The unbalanced sex also reflects the situation for hospital in Taiwan. We think the unbalanced sex could interfere the results of analysis, therefore, we adopted multiple linear regression to simultaneously control possible interference factors from marriage state, parenthood, exercise habit, alcohol use habit, education degree, shift work, and medical profession. Despite this, we still ignore culture factors that gender inequality exists in Taiwan society. That should be further explored in future study of burnout.

The present study did not include the work loading on individuals that could lead to interfere the sex effect on burnout despite we had controlled professional factor. The future research should adopt suitable work loading scale to measure this possible risk factor. Notably, healthcare workers could face potential crisis on medical or occupational violence that could lead to increased burnout. Therefore, the future research for burnout also should consider this issue.

Due to the mediation model of an observational study could be biased [62], the causal relationship among women, NBSP, and burnout is higher risk of judgement. Therefore, we avoid using the sentence “causal relationship” in our conclusion. Despite this, indirect effect (women-NBSP-PB/WB) is significantly strong than direct effect (women-PB/WB) in statistic, that represents NBSP still play a key role for women effect on burnout.

## 5. Conclusions

Neck and shoulder pain is commonly associated with burnout among healthcare workers, with a higher prevalence among women than among men. With confounders being controlled for, women were found to experience substantially higher levels of personal and work-related burnout than men did. Notably, neck and shoulder pain are a key reason why women are more likely than men to experience burnout, a factor which has rarely been mentioned in the literature. Therefore, to mitigate the symptoms of burnout among women, medical institutions should adopt strategies that are tailored to women, such as targeting musculoskeletal pain through interventions, to prevent burnout among them.

## Figures and Tables

**Table 1 ijerph-19-12872-t001:** MS pain sites and factor analysis of the Nordic musculoskeletal questionnaire.

MS Pain Site	MS Pain Participants	Prevalence %	Frequency Score	Factor Loading		
			Mean ± SD	Factor 1	Factor 2	Factor 3
Neck	585	36.22	26.76 ± 37.64	0.33	−0.02	−0.03
Left shoulder	325	20.12	15.07 ± 31.62	0.33	−0.01	−0.01
Right shoulder	371	22.97	17.64 ± 33.89	0.33	0.02	−0.07
Upper back	273	16.90	12.90 ± 29.77	0.17	0.00	−0.01
Waist or lower back	451	27.93	20.20 ± 34.72	0.08	−0.04	0.03
Left elbow	70	4.33	3.29 ± 16.26	−0.05	−0.04	−0.05
Right elbow	113	7.00	5.33 ± 20.43	−0.04	−0.04	−0.02
Left wrist	77	4.77	3.72 ± 17.38	−0.05	0.00	0.01
Right wrist	162	10.03	7.51 ± 23.66	−0.03	−0.03	−0.02
Left hip/thigh/buttock	67	4.15	3.12 ± 15.64	−0.05	−0.07	−0.01
Right hip/thigh/buttock	68	4.21	3.17 ± 15.83	−0.02	−0.04	−0.06
Left knee	80	4.95	3.78 ± 16.98	−0.05	−0.07	0.51
Right knee	88	5.45	4.17 ± 18.05	−0.02	−0.04	0.45
Left ankle	29	1.80	1.26 ± 10.10	−0.02	0.49	−0.05
Right ankle	25	1.55	1.10 ± 9.58	−0.02	0.54	−0.05
		eigenvalues		4.93	1.55	0.68
		explained variation %		57.59	18.12	0.08

**Table 2 ijerph-19-12872-t002:** Stratified analysis for sex and burnout.

Survey Variables	Subjects	Subjects (%)			PB		WB	
		Men	Women	*p*	Mean ± SD	*P*	Mean ± SD	*p*
*Marriage state*								
Married	779	169 (56.15)	610 (46.42)	0.003 ^†^	35.54 ± 17.68	0.240	31.99 ± 15.38	<0.0001
Unmarried	836	132 (43.85)	704 (53.58)		36.60 ± 18.38		36.29 ± 16.76	
*Parenthood*								
Yes	703	155 (51.50)	548 (41.70)	0.002 ^†^	35.49 ± 18.51	0.243	31.47 ± 15.83	<0.0001
No	912	146 (48.50)	766 (58.30)		36.55 ± 17.68		36.33 ± 16.25	
*Exercise habit weekly*								
regular exercise weekly	933	211 (70.10)	722 (54.95)	<0.0001 ^†^	33.43 ± 17.18	<0.0001	31.85 ± 15.95	<0.0001
None regular exercise weekly	682	90 (29.90)	592 (45.05)		39.73 ± 18.59		37.45 ± 16.10	
*Alcohol use habit in a month*								
Ever alcohol use	609	157 (52.16)	452 (34.40)	<0.0001 ^†^	38.47 ± 17.68	<0.0001	36.40 ± 15.67	<0.0001
Never alcohol use	1006	144 (47.84)	862 (65.60)		34.65 ± 18.13		32.89 ± 16.45	
*Sleeping duration (per day) ranks*								
<6 h	626	118 (39.20)	508 (38.66)	0.896 ^†^	41.07 ± 18.84		38.05 ± 17.01	<0.0001
>6 h	989	183 (60.80)	806 (61.34)		32.94 ± 16.79		31.78 ± 15.26	
*Education degree*								
Master’s degree or above	297	86 (28.57)	211 (16.06)	<0.0001 ^†^	36.15 ± 34.08	0.946	32.41 ± 15.79	0.034
University or below university degree	1318	215 (71.43)	1103 (83.94)		36.07 ± 18.03		34.62 ± 16.33	
*Overtime work per month*								
Experience overtime work	561	112 (37.21)	449 (34.17)	0.315 ^†^	42.40 ± 18.09	<0.0001	40.01 ± 15.71	<0.0001
Seldom overtime work	1054	189 (62.79)	865 (65.83)		32.73 ± 17.11		31.13 ± 15.68	
*Shift schedules*								
IRS work	192	33 (10.96)	159 (12.10)	0.029	42.71 ± 18.58	<0.0001 ^§^	40.94 ± 16.50	<0.0001 ^§^
RS work	196	24 (7.97)	172 (13.09)		39.44 ± 19.21		38.30 ± 16.73	
Night shift work	166	27 (8.97)	139 (10.58)		36.32 ± 18.96		35.31 ± 16.45	
DS work	1061	217 (72.09)	844 (64.23)		34.24 ± 17.22		32.07 ± 15.59	
*Profession*								
Physicians	138	92 (30.56)	46 (3.50)	<0.0001	41.91 ± 20.15	<0.0001 ^§^	39.67 ± 17.34	<0.0001 ^§^
Nurses	613	26 (8.64)	587 (44.67)		40.37 ± 18.11		38.23 ± 16.26	
PTs	283	70 (23.26)	213 (16.21)		33.80 ± 17.01		31.95 ± 15.64	
ADs	581	113 (37.54)	468 (35.62)		31.30 ± 16.47		29.78 ± 14.82	
*Presence of chronic diseases*								
Yes	638	130 (43.19)	508 (38.66)	0.151 ^†^	38.92 ± 18.14	<0.0001	36.31 ± 16.57	<0.0001
No	977	171 (56.81)	806 (61.34)		34.24 ± 17.75		32.84 ± 15.89	

*p*, *p* value; ^†^, Fisher exact test; SD, standard deviation; ^§^, ANOVA.

**Table 3 ijerph-19-12872-t003:** Association of survey variables with sex/burnout.

Survey Variable	Men	Women		PB		WB	
	Mean ± SD	Mean ± SD	*p*	β	*p*	β	*p*
LAFF	57.06 ± 21.84	56.75 ± 20.03	0.824	−0.12	<0.0001	−0.14	<0.0001
Work experience	10.58 ± 9.03	10.72 ± 9.59	0.823	−0.17	0.001	−0.23	<0.0001
NBSP	−0.17 ± 0.84	0.04 ± 0.93	0.000	8.25	<0.0001	6.32	<0.0001
BAP	0.02 ± 0.90	−0.01 ± 0.85	0.643	1.46	0.006	1.42	0.003
BKP	−0.04 ± 0.70	0.01 ± 0.78	0.321	1.98	0.001	1.31	0.013

*p*, *p* value; β, linear regression coefficient of dummy variable women against PB/WB when added survey variable; SD, standard deviation.

**Table 4 ijerph-19-12872-t004:** Effect of being a woman on burnout in linear regression models.

Models	PB			WB		
	β	*p*	adj. R^2^	Β	*P*	adj. R^2^
M_0_	1.84	0.111	0.00	1.80	0.083	0.00
M_1_	2.56	0.030	0.18	2.52	0.017	0.18
M_2_	2.83	0.017	0.18	2.43	0.022	0.18
M_3_	2.86	0.016	0.18	2.34	0.027	0.19
M_4_	2.56	0.029	0.18	2.54	0.016	0.19
M_5_	0.70	0.522	0.30	1.12	0.263	0.27

*p*, *p* value; β, linear regression coefficient effect of being a woman; M_0_, without adjusted variables; M_1_, adjusting for weekly exercise habits, monthly alcohol use, sleep duration per day, level of education, overtime work per month, shift schedules, profession, presence of presence of chronic diseases, LAFF, work experience; M_2_, M_1_ + marriage; M_3_, M_1_ + parenthood; M_4_, M_1_ + BAP and BKP; M_5_, M_4_ + NBSP.

**Table 5 ijerph-19-12872-t005:** Mediation effect of NBSP between sex and burnout.

		Mediating Factor (NBSP)					
DV	c	*c*′	*a*	*s_a_*	*b*	*s_b_*	Z
PB	1.84	0.08	0.21 **	0.06	8.25 ***	0.45	3.44 **
WB	1.80	0.46	0.21 **	0.06	6.30 ***	0.41	3.41 **

** *p* < 0.01; *** *p* < 0.0001; *a*, the linear regression coefficient for women dummy variable against NBSP in first stage effect; *s_a_*, the standard error for *a*; *b*, the linear regression coefficient of mediating factor NBSP against PB and WB in second stage effect; *s_b_*, the standard error for *b*.

## Data Availability

The datasets used and/or analyzed during the current study are available from the corresponding author on reasonable request.

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
