# Peer review of "Mediation Effect of Musculoskeletal Pain on Burnout: Sex-Related Differences"

_ijerph, 2022, doi:10.3390/ijerph191912872_

Round 1
Reviewer 1 Report
Interesting paper and detailed study with sophisticated methodology. I recommend adding more to your discussion to make this applicable to your reader. Consider further recommendations/suggestions on alleviating burnout through addressing/preventing pain or ergonomics to practitioners, stakeholders, and employers.
Author Response
Reviewer 1:
Suggestions for revision 1: Interesting paper and detailed study with sophisticated methodology. I recommend adding more to your discussion to make this applicable to your reader.
Response 1:
Thanks to the reviewer for the suggestion. We have added new paragraphs guiding readers to clearly understand some analysis methods and results. such as:
Line 158-170:
Regarding the procedures of statistical analyses, we adopted four steps to deter-mined women's effect on burnout and their mediating factors.
Step 1: Factor analysis [37] was conducted on the Nordic Musculoskeletal Questionnaire results to determine the underlying variables that explained most of the responses.
Step 2, The present research adopted the chi-square test/Fisher exact test (categorical variables) and t-test or one-way ANOVA (continuous variables) to determine if the difference between survey variable and sex or PB/WB is significant in statistics. These survey variables were significant in statistics that would be possible mediating factors between sex and burnout or risk/protective factors of burnout.
Step 3: The risk or protective factors of burnout would be added in linear regression models of women's effect on burnout to determine whether controlling for covariates significantly affected associations of the independent variables (IVs) with the dependent variable (DV).
Step 4: The mediation effect would be tested on…………..
Line 203-204:
First, the research must determine the underlying variables that could maximum explain the Nordic Musculoskeletal Questionnaire.
Line 218-219:
Second, the present study would determine the sex-related difference in survey variables for determining possible mediating factors.
Line 230-231:
Confounders of burnout were next identified and used as control variables in a multiple linear regression model of women's effect on burnout.
Line 269-270:
Mediation effect of NBSP that women sustain high burnout would be further identified by mediation models.
Line 390-392: According to the two results, we could determine women more easily suffer from neck and both shoulders pain than men, which would be an important reason that women sustain high PB and WB than men.
Line 403-404:
Despite this, we still ignore cultural factors that gender inequality exists in Taiwanese society. That should be further explored in future studies of burnout.
Line 408-410:
Notably, healthcare workers could face potential crises on medical or occupational violence that could lead to increased burnout. Therefore, future research on burnout also should consider this issue.
Suggestions for revision 2: Consider further recommendations/suggestions on alleviating burnout through addressing/preventing pain or ergonomics to practitioners, stakeholders, and employers.
Response 2:
Thanks to the reviewer for the suggestion. According to our findings, we suggest in “Conclusion”.
Line 422-425:
Therefore, to mitigate the symptoms of burnout among women, medical institutions should adopt strategies that are tailored to women, such as targeting musculoskeletal pain through interventions, to prevent burnout among them.

Reviewer 2 Report
Thank you for the opportunity to review this article. The time involved in submitting your manuscript is greatly appreciated.
Despite this, the article presents a series of issues that must be noted and mended. The recommendations are presented separately in sections. Hopefully, they would be useful.
Title: the title does not adequately reflect the content of the paper. Please, try to change it to better inform the readers about the relationships between the variables that you test and also inform them about the quality of your sample.
Introduction:
Less information appears in the introduction. More information would be better. Maybe expanded by adding the most relevant theories or theoretical models that explain burnout occurrence.
Moreover, some of the references that you cite are too old. Even though the most relevant studies should be referenced, also the RECENT research must be included.
At the end of the literature review, the aims and the questions in the research should appear. Maybe formulating the questions as a hypothesis would be an option to clear this aspect. Another commentary is the possibility of including this part in the final of the introduction part; even a separate section could be a good option, in order to clear the final of the introduction and to serve as a connection with the method.
Method:
Please, try to better describe the sociodemographic data of your participants. In the same sense, give the readers with detailed information about the procedure for recruiting participants and collecting data.
The response options could also be reported by the minimum and the maximum, (i. e.: 100= Always, 0= Never), without detailing any possible response.
Related to the instruments, please better inform about their psychometric quality and give the readers some examples of the items. If you can, please inform me about previous studies where the same instrument has been used and the reliability obtained in that research.
Data analyses
Please, explain to the readers which procedures of statistical analyses have been used and justify your decisions.
Results
The results should be presented in the same order as the introduction and hypotheses. Also, the same order must be used in the Tables. This simplifies the work for readers.
Finally, the repetition is constant all over the article. Please, try to change the words in order to do the reading more interesting and motivating.
Discussion:
First of all, try to better adjust your conclusions to the findings. Or to say in other words, please try to justify more clearly the connection between your conclusions and your findings.
Finally, a section related to limitations, future lines of investigation, and the principal contributions of the research could be attractive. Your paper has a lot of relevant implications for society and policymakers, but you need to elaborate more on this topic.
Conclusion:
They don’t appear to new conclusions on this part. This part does not add any new to the rest of the paper. Please, try to condense your findings, or highlight your main contribution to the field.
Minor point: APA recommends avoiding a lot of abbreviations because it is very difficult for readers to keep in mind all of these abbreviations.
Author Response
Reviewer 2
Thank you for the opportunity to review this article. The time involved in submitting your manuscript is greatly appreciated.
Despite this, the article presents a series of issues that must be noted and mended. The recommendations are presented separately in sections. Hopefully, they would be useful.
Suggestions for revision 1: Title: the title does not adequately reflect the content of the paper. Please, try to change it to better inform the readers about the relationships between the variables that you test and also inform them about the quality of your sample.
Response 1:
Thanks to the reviewer for the correction. We have rewritten the original Title as “Mediation effect of musculoskeletal pain on burnout: Sex-related differences”
Introduction:
Suggestions for revision 2-1: Less information appears in the introduction. More information would be better. Maybe expanded by adding the most relevant theories or theoretical models that explain burnout occurrence.
Response 2-1
Thanks to the reviewer for the correction. We have a new paragraph and reference [2] to strengthen the explanation of the burnout model.
Line 32-41: The development of burnout can be divided into five stages[2], the first stage is the enthusiasm period, people kept enthusiasm at work. The second stage is the so-called stagnation period, people begin to suffer from a series of work pressures causing decreasing enthusiasm, the burnout risks become initiated. The third stage is the frustration period, beginning to gradually form the so-called Chronic stress causes people to gradually lose their enthusiasm for work. The fourth stage is the apathy stage, which resulted without obvious returns and work powerlessness. The fifth stage is the intervention period, at which time habitual burnout will lead to physical and Emotional problems, and people will start to seek help and intervention.
Suggestions for revision 2-2: Moreover, some of the references that you cite are too old. Even though the most relevant studies should be referenced, also the RECENT research must be included.
Response 2-2:
Reference 4 “4. Schaufeli, W.B., et al., On the clinical validity of the Maslach burnout inventory and the burnout measure. Psychology & Health, 2001. 16(5): p. 565-582.” was renewed as “Gavelin, H.M., et al., Cognitive function in clinical burnout: A systematic review and meta-analysis. Work & Stress, 2022. 36(1): p. 86-104.”
Reference 26 “Langballe, E.M., et al., The relationship between burnout and musculoskeletal pain in seven Norwegian occupational groups. Work, 2009. 32(2): p. 179-188.” was renewed as “Engelbrecht, G.J., L.T. de Beer, and W.B. Schaufeli, The relationships between work intensity, workaholism, burnout, and self-reported musculoskeletal complaints. Human Factors and Ergonomics in Manufacturing & Service Industries, 2020. 30(1): p. 59-70.”
Suggestions for revision 2-3: At the end of the literature review, the aims and the questions in the research should appear. Maybe formulating the questions as a hypothesis would be an option to clear this aspect. Another commentary is the possibility of including this part in the final of the introduction part; even a separate section could be a good option, in order to clear the final of the introduction and to serve as a connection with the method.
Response 2-3: Thanks to the reviewer for the correction. We had renewed the original paragraph as a new paragraph in lines 74-79.
Line 65-70: Based on the associations among women, MS pain, and burnout, whether women sustain high burnout than men due to MS pain was a noteworthy and interesting topic. Therefore, this study proposed two hypotheses to further explore: that sex differences in burnout (1) exist and (2) result from MS pain differences between women and men. Hoping the results of the research could encourage medical institutions to make sex-differentiated strategies for burnout to further reduce burnout, especially among female staff members.
Method:
Suggestions for revision 3-1: Please, try to better describe the sociodemographic data of your participants. In the same sense, give the readers with detailed information about the procedure for recruiting participants and collecting data.
Response 3-1: Thanks to the reviewer for the correction. We have added a new paragraph to further descript the participants. In addition, the procedure for recruiting participants and collecting data adopted google forms. These detailed descriptions were below:
Line 76-79: The professional fields and sex of 1615 participants were in Supplementary Information Table S1, those reclassified as Physicians, Nurses, Professional and technical personnel, and Administration staff.
Line 74-75: …..QR code of “google forms”….
Suggestions for revision 3-2: The response options could also be reported by the minimum and the maximum, (i. e.: 100= Always, 0= Never), without detailing any possible response.
Response 3-2:
Thanks to the reviewer for the correction. We have rewritten the paragraph in Line 154-155 “…., which was inversely scored (i.e., the responses were scored by minimum "always"= 0 and maximum "never/almost never"= 100, sequentially)”.
Suggestions for revision 3-3: Related to the instruments, please better inform about their psychometric quality and give the readers some examples of the items. If you can, please inform me about previous studies where the same instrument has been used and the reliability obtained in that research.
Response 3-3:
We adopted the NMQ to measure the MS pain. According to the study of a random sample of 1201 participants in the fourth wave Health Survey in 2022, the reliability of NMQ was good (kappa values were between 0.51-0.68). Therefore, we added a sentence and reference in Line 113-114. “and the reliability was good (kappa values were between 0.51-0.68) [31]”.
In addition, we adopted CBI to measure burnout levels. Three measurement scales were high internal reliability who’s the Cronbach’s alphas were between 0.85 and 0.87. Therefore, we added a sentence in Line 132-133. “…. had very high internal reliability (the Cronbach’s alphas were between 0.85 and 0.87) and”…..
Data analyses
Suggestions for revisions 3-4: Please, explain to the readers which procedures of statistical analyses have been used and justify your decisions.
Response 3-4:
We have rewritten and added new paragraphs to strengthen the explanation of the procedures of statistical analyses.
Line 158-171: Regarding the procedures of statistical analyses, we adopted four steps to determine women's effect on burnout and their mediation factors.
Step 1: Factor analysis [37] was conducted on the NMQ results to determine the underlying variables that explained most of the responses.
Step 2, The present research adopted the chi-square test/Fisher exact test (categorical variables) and t-test or one-way ANOVA (continuous variables) to determine if the difference between survey variable and sex or PB/WB is significant in statistics. These survey variables were significant in statistics that would be possible mediation factors between sex and burnout or risk/protective factors of burnout.
Step 3: The risk or protective factors of burnout would be added in linear regression models of women's effect on burnout to determine whether controlling for covariates significantly affected associations of the independent variables (IVs) with the dependent variable (DV).
Step 4: The mediation effect would be tested on…….
Results
Suggestions for revision 4:
The results should be presented in the same order as the introduction and hypotheses. Also, the same order must be used in the Tables. This simplifies the work for readers.
Response 4:
Thanks to the reviewer for the correction. However, we think Factor analysis should be adopted first because we must determine the underlying variables NBSP, BAP, and BKP that explained most of the responses. They could help us to determine the association between MS pain and burnout. To help the reader, we descript the procedure of data analysis through steps 1, 2, 3, and 4 in “Methods”. Tables 1- 5 could correspond order of steps 1-4. Therefore, we hope the reviewer could agree to the original order of Tables 1-5.
Line 158-171:
Regarding the procedures of statistical analyses, we adopted four steps to deter-mined women's effect on burnout and their mediating factors.
Step 1: Factor analysis [37] was conducted on the Nordic Musculoskeletal Questionnaire results to determine the underlying variables that explained most of the responses.
Step 2, The present research adopted the chi-square test/Fisher exact test (categorical variables) and t-test or one-way ANOVA (continuous variables) to determine if the difference between survey variable and sex or PB/WB is significant in statistics. These survey variables were significant in statistics that would be possible mediating factors between sex and burnout or risk/protective factors of burnout.
Step 3: The risk or protective factors of burnout would be added in linear regression models of women's effect on burnout to determine whether controlling for covariates significantly affected associations of the independent variables (IVs) with the dependent variable (DV).
Step 4: The mediation effect would be tested on…………..
Suggestions for revision 5:
Finally, the repetition is constant all over the article. Please, try to change the words in order to do the reading more interesting and motivating.
Response 5: Thanks to the reviewer for the suggestion. We added new sentences to clearly introduced the purpose of every paragraph and Table.
Line 203-204: First, the research must determine the underlying variables that could maximum explain NMQ.
Line 218-219: Second, the present study would determine the sex-related difference in survey variables for determining possible mediating factors.
Line 230-231: Confounders of burnout were next identified and used as control variables in a multiple linear regression model of women's effect on burnout.
Line 269-270: The mediation effect of NBSP that women sustain high burnout would be further identified by mediation models.
Discussion:
Suggestions for revision 6:
First of all, try to better adjust your conclusions to the findings. Or to say in other words, please try to justify more clearly the connection between your conclusions and your findings.
Response 6:
Thanks to the reviewer for the correction. We added a new paragraph to clearly connect our findings.
Line 390-392: According to the two results, we could determine women more easily suffer from neck and both shoulders pain than men, that would be an important reason that women sustain high PB and WB than men.
Suggestions for revision 7:
Finally, a section related to limitations, future lines of investigation, and the principal contributions of the research could be attractive. Your paper has a lot of relevant implications for society and policymakers, but you need to elaborate more on this topic.
Response 7:
Thanks to the reviewer for the correction. We added two paragraphs to elaborate on the topic.
Line 403-404: Despite this, we still ignore cultural factors that gender inequality exists in Taiwanese society. That should be further explored in future studies of burnout.
Line 408-410: Notably, healthcare workers could face a potential crisis on medical or occupational violence that could lead to increased burnout. Therefore, future research on burnout also should consider this issue.
Conclusion:
Suggestions for revision 8:
They don’t appear to new conclusions on this part. This part does not add any new to the rest of the paper. Please, try to condense your findings, or highlight your main contribution to the field.
Response 8:
Thanks to the reviewer for the correction. We have rewritten our Conclusion, detailed content follows:
Neck and shoulder pain is commonly associated with burnout among healthcare workers, with a higher prevalence among women than among men. With confounders being controlled for, women were found to experience substantially higher levels of personal and work-related burnout than men did. Notably, neck and shoulder pain is a key reason why women are more likely than men to experience burnout, a factor that has rarely been mentioned in the literature. Therefore, to mitigate the symptoms of burnout among women, medical institutions should adopt strategies that are tailored to women, such as targeting musculoskeletal pain through interventions, to prevent burnout among them.
Suggestions for revision 9:
Minor point: APA recommends avoiding a lot of abbreviations because it is very difficult for readers to keep in mind all of these abbreviations.
Response 9:
Thanks to the reviewer for the correction. We have renewed our Abbreviations detailed follow:
PB: personal burnout; WB: work-related burnout; MS: musculoskeletal; LAFF: leisure activities with family or friends; NBSP: neck and both shoulders pain; BAP: both ankles pain; BKP: both knees pain; IV: independent variable; DV: dependent variable.

Reviewer 3 Report
The authors presented relevant empirical research about burnout in the health sector. The article could be interesting for the reader but some improvements could be made:
a) Abstract could be shortened leaving main notes to achieved results but deleting specifically counted numbers and similar results.
b) The authors give a hypothesis but it is not clear what the goal is. The introduction could be improved by expanding the detalization of the scientific problem, identifying and describing the scientific goal.
c) The article is missing the literature review part. The authors cited around 60 references in the article but the newest research on the topic is not analyzed (less than 20 percent of cited references could be assumed as the newest). Even more, the authors mention a literature review on the 353rd line of the article but such a part does not exist.
d) The discussion part is sufficient and explicit.
e) Conclusions could be improved by enclosing more profound results of empirical research and its impact on science and practice.
Author Response
The authors presented relevant empirical research about burnout in the health sector. The article could be interesting for the reader but some improvements could be made:
Suggestions for revision 1:
- a) Abstract could be shortened leaving main notes to achieved results but deleting specifically counted numbers and similar results.
Response 1:
Thanks to the reviewer for the correction. We fixed my abstract detailed following:
Line 13-27: Burnout occurs when people are exposed to emotionally demanding work situations over an extended period, resulting in physical, emotional, and mental exhaustion. This study adopted the personal burnout (PB) and work-related burnout (WB) scales of the Copenhagen Burnout Inventory to measure burnout levels. Musculoskeletal (MS) pain is one-factor influencing burnout. Previous studies have demonstrated that male and female hormones may contribute to sex-related differences in MS pain. This was an observational and cross-sectional study conducted at a medical-university-affiliated hospital in Taichung, Taiwan, in 2021. Data were collected for demographic characteristics, family structure, living habits, occupation, physical health, Nordic Musculoskeletal Questionnaire score, and Copenhagen Burnout Inventory score. Of the 2,531 questionnaires that were distributed, 1,615 (63.81%) valid questionnaires remained after those with missing data were excluded. The results demonstrated neck and shoulder pain (NBSP) is commonly associated with burnout among healthcare workers, with a higher prevalence among women than among men. With confounders being controlled for, women were found to experience substantially higher levels of personal and work-related burnout than men did. NBSP is a mediating factor, therefore, it is a key reason why women are more likely than men to experience burnout.
Suggestions for revision 2:
- b) The authors give a hypothesis but it is not clear what the goal is. The introduction could be improved by expanding the detail of the scientific problem, identifying and describing the scientific goal.
Response 2:
Thanks to the reviewer for the correction. We added and rewritten two paragraphs to improve the problem.
Line 65-66: Based on the associations among women, MS pain, and burnout, whether women sustain high burnout than men due to MS pain was a noteworthy and interesting topic.
Line 68-70: Hoping the results of the research could encourage the medical institutions to make sex-differentiated strategies for burnout to further reduce burnout, especially among female staff members.
Suggestions for revision 3: c) The article is missing the literature review part. The authors cited around 60 references in the article but the newest research on the topic is not analyzed (less than 20 percent of cited references could be assumed as the newest).
Response 3: Thanks to the reviewer for the correction. We have renewed our references detailed following:
- De Hert, S., Burnout in Healthcare Workers: Prevalence, Impact and Preventative Strategies. Local Reg Anesth, 2020. 13: p. 171-183.
- Gavelin, H.M., et al., Cognitive function in clinical burnout: A systematic review and meta-analysis. Work & Stress, 2022. 36(1): p. 86-104.
- Yoshimoto, T., et al., The Economic Burden of Lost Productivity due to Presenteeism Caused by Health Conditions Among Workers in Japan. J Occup Environ Med, 2020. 62(10): p. 883-888.
- Engelbrecht, G.J., L.T. de Beer, and W.B. Schaufeli, The relationships between work intensity, workaholism, burnout, and self-reported musculoskeletal complaints. Human Factors and Ergonomics in Manufacturing & Service Industries, 2020. 30(1): p. 59-70.
- Aulia, C., VALIDITY AND RELIABILITY TEST OF THE NORDIC MUSCULOSKELETAL QUESTIONNAIRE WITH FORMAL AND INFORMAL SECTOR WORKERS, in 7th International Conference on Public Health 2020. 2020: Indonesia. p. 100-106.
- Dahl, A.G., S. Havang, and K. Hagen, Reliability of a self-administrated musculoskeletal questionnaire: The fourth Trøndelag health study. Musculoskeletal Science and Practice, 2022. 57: p. 102496.
- Strikwerda, M., et al., The Association of Burnout and Vital Exhaustion With Type 2 Diabetes: A Systematic Review and Meta-Analysis. Psychosomatic Medicine, 2021. 83(9).
- Roslan, N.S.; Yusoff, M.S.B.; Asrenee, A.R.; Morgan, K. Burnout Prevalence and Its Associated Factors among Malaysian Healthcare Workers during COVID-19 Pandemic: An Embedded Mixed-Method Study. Healthcare 2021; 9(1):90.
- Du, J., et al., Relationship Between the Exposure to Occupation-related Psychosocial and Physical Exertion and Upper Body Musculoskeletal Diseases in Hospital Nurses: A Systematic Review and Meta-analysis. Asian Nursing Research, 2021. 15(3): p. 163-173.
Suggestions for revision 4: Even more, the authors mention a literature review on the 353rd line of the article but such a part does not exist.
Response 4: Thanks to the reviewer for the correction. We deleted the original sentence “According to the above results and literature review, sex difference really exists on NBSP and burnout, too.” and added a new paragraph (Line 379-380) “Based on the above analysis, sex difference really exists on NBSP and burnout.” Therefore, the new paragraph details follow:
Line (376-382): Table 3 presents data showing that women sustained higher levels of NBSP than men (women vs. men: 0.04 ± 0.93 vs. −0.17 ± 0.84; p < .01). In addition, M1, M2, M3, and M4 models in Table 4 have determined women really sustain high level on burnout than men in adjusting confounders. Based on the above analysis, sex difference really exists in NBSP and burnout. To explore the NBSP effect on the relationship between sex and burnout, we added NBSP to the multiple linear regression model in Table 4 (M4) and as a new model M5.
Suggestions for revision 5: d) The discussion part is sufficient and explicit.
Response 5: Thanks for the reviewer's affirmation
Suggestions for revision 6: e) Conclusions could be improved by enclosing more profound results of empirical research and its impact on science and practice.
Response 7: Thanks to the reviewer for the correction. We have rewritten my Conclusions detailed follow:
Neck and shoulder pain is commonly associated with burnout among healthcare workers, with a higher prevalence among women than among men. With confounders being controlled for, women were found to experience substantially higher levels of personal and work-related burnout than men did. Notably, neck and shoulder pain is a key reason why women are more likely than men to experience burnout, a factor that has rarely been mentioned in the literature. Therefore, to mitigate the symptoms of burnout among women, medical institutions should adopt strategies that are tailored to women, such as targeting musculoskeletal pain through interventions, to prevent burnout among them.
